# Clinical Associations between Serial Electrocardiography Measurements and Sudden Cardiac Death in Patients with End-Stage Renal Disease Undergoing Hemodialysis

**DOI:** 10.3390/jcm10091933

**Published:** 2021-04-29

**Authors:** Hyun Jin Lee, A Reum Choe, HaeJu Lee, Dong Ryeol Ryu, Ea Wha Kang, Jung Tak Park, Su Hwan Lee, Junbeom Park

**Affiliations:** 1Department of Internal Medicine, Severance Hospital, Yonsei University College of Medicine, Seoul 03722, Korea; HJLEE91@yuhs.ac; 2Department of Internal Medicine, College of Medicine, Ewha Womans University, Seoul 07985, Korea; prems0121@naver.com (A.R.C.); drryu@ewha.ac.kr (D.R.R.); 3Department of Thoracic and Cardiovascular Surgery, Seoul National University Hospital, Seoul 03080, Korea; 2haejude@gmail.com; 4Division of Nephrology, Department of Internal Medicine, National Health Insurance Service Ilsan Hospital, Goyang 10444, Korea; eawha@nhimc.or.kr; 5Department of Internal Medicine, College of Medicine, Institute of Kidney Disease Research, Yonsei University, Seoul 03722, Korea; JTPARK@yuhs.ac; 6Division of Pulmonary and Critical Care Medicine, Department of Internal Medicine, Severance Hospital, Yonsei University College of Medicine, Seoul 03722, Korea; 7Division of Cardiology, Department of Internal Medicine, College of Medicine, Ewha Womans University, Seoul 07985, Korea

**Keywords:** critical care, hemodialysis, death, sudden, cardiac, electrocardiography, kidney failure, arrhythmias

## Abstract

The rate of sudden cardiac death (SCD) for hemodialysis (HD) patients is significantly higher than that observed in the general population and have the highest risk for arrhythmogenic death. In this multi-center study, patients starting hemodialysis in each hospital were enrolled; they underwent regular check-ups in an open-patient clinic. We examined serial electrocardiography (ECG) data in patients undergoing HD and determined their associations with the occurrence of SCD. Of 678 enrolled subjects who underwent serial ECG before and after hemodialysis, 291 died and 39 developed SCD. In all subjects, the QT peak-to-end (QTpe) interval at all leads and QRS duration were shortened after hemodialysis. The SCD group showed a significant change in the QTpe interval of the inferior, anterior, and lateral leads before and after hemodialysis compared with the survivor group (*p* < 0.001). In the pre-hemodialysis ECG, SCD patients had significantly longer QTpe intervals in all leads (*p* < 0.001) and a longer QRS duration (92.6 ± 14.0 vs. 100.6 ± 14.9 ms, *p* = 0.015) than survivors. In conclusion, patients with a longer QTpe interval before hemodialysis and large changes in ECG parameters after hemodialysis might be at a higher risk of SCD. Therefore, changes in the ECG before and after hemodialysis could help to predict SCD.

## 1. Introduction

According to the 2016 United States Renal Data System database, the most common cause of mortality in patients with end-stage kidney disease (ESRD) undergoing dialysis is cardiovascular disease (CVD), accounting for 41% of all deaths and 54% of deaths with known causes in this population [1]. Among CVD deaths, sudden cardiac death (SCD) is the largest cause of mortality in ESRD patients undergoing hemodialysis (HD) [2]. Although patients with ESRD undergoing HD constitute a relatively small population, SCD is significantly more common among them than in the general population, and ESRD is the strongest risk factor for arrhythmogenic death [3]. Several studies have identified risk factors for SCD in patients with ESRD undergoing HD, such as myocardial hypertrophy [4], left ventricular diastolic dysfunction [5], myocardial fibrosis [6], and dialysis-induced myocardial ischemia [7]. Furthermore, during HD, wide fluctuation of electrolytes such as sodium, potassium, calcium, magnesium [8], and other ions may increase arrhythmia risk in the long term as well as during HD [9,10].

Therefore, if patients at a high risk of SCD could be identified, preventive management could be provided, including drug therapy and implantable cardioverter-defibrillator placement. However, invasive methods such as coronary angiography and treadmill stress tests are required to predict SCD risk in patients undergoing HD.

Among non-invasive methods, electrocardiography (ECG) reflects myocardial changes such as the prolongation of QT interval, QTc interval, QTc dispersion, and QT peak-to-end (QTpe) interval [11,12,13,14,15]. As mentioned in many previous reports, several risk factors of SCD cause ventricular remodeling and myocardial fibrosis, leading to ventricular dysfunction and dys-synchronization [5,6,7], changes which are known to affect SCD risk [11,12,13]. Notwithstanding this accumulating evidence, studies in large populations are still lacking, because of the low prevalence of ESRD [1,11]. In this study, we used a multi-center database to examine (1) the serial changes in ECG findings in patients undergoing HD and (2) how baseline ECG measurements and their changes are related to the occurrence of cardiovascular diseases and SCD.

## 2. Methods

### 2.1. Study Design and Population

This multi-center, retrospective cohort study was approved by the Institutional Review Board of the Ewha Womans University Mokdong hospital (2017-08-005) and conducted in three tertiary referral hospitals (Yonsei University Medical Center, Ewha Womans University Medical Center, and National Health Insurance Service Ilsan Hospital) in South Korea. This study enrolled patients who started HD in each hospital from November 1986 to September 2015. In addition, the last follow up date was 26 November 2016. The exclusion criteria were moderate to severe valvular heart diseases, congenital heart diseases, acute myocardial infarction/unstable angina within the past 3 months, and any uncontrolled metabolic disorders such as homocystinuria, glycogen storage diseases, and mitochondrial disorders. The rate of SCD and ECG measurements before and after HD were analyzed.

### 2.2. Data Collection

Data from all subjects were obtained through review of the hospital’s medical records. In case of missing medical records about the cause of death, telephone interviews were conducted. Clinical data such as those of demographic characteristics, start date of HD, presence of hypertension and diabetes mellitus, cause of death, and mortality were evaluated. Various laboratory data such as those of complete blood counts and blood urea nitrogen, creatinine, and electrolyte levels after HD were collected. All subjects underwent 12-lead ECG and 2-dimensional transthoracic echocardiography (TTE) before starting HD, which were repeated at regular intervals, together with laboratory tests, after starting HD. Furthermore, cardiac function, in terms of ejection fraction (EF), left ventricle end-diastolic dimension, left ventricle end-systolic dimension, and E/E’ ratio (E: mitral peak velocity of early filling, E’: early diastolic mitral annular velocity), was measured by TTE before and after HD.

### 2.3. Definitions

According to the 2006 American College of Cardiology/American Heart Association/Heart Rhythm Society guidelines, SCD is defined as the sudden cessation of cardiac activity regardless of whether resuscitation or spontaneous reversion occurs [16]. The most commonly used definition of SCD is “death occurring within an hour of symptom onset with no clinical support for another cause,” [17] and we used the same definition in our study.

Cause of death was defined as the event which directly led to death and was divided into the following six mutually exclusive categories: (1) cardiovascular disease (including SCD), (2) intracerebral hemorrhage, (3) major bleeding, (4) cancer, (5) sepsis/other infection, or (6) other. Among them, major bleeding was defined as (1) fatal bleeding; (2) symptomatic bleeding in a critical area or organ, such as intraspinal, intraocular, retroperitoneal, intraarticular, pericardial, or intramuscular bleeding with compartment syndrome; and/or (3) bleeding causing a decrease in the hemoglobin level of ≥2 g/dL or leading to the requirement of transfusion. When the cause of death did not belong to the first five categories, deaths were classified as deaths due to “other” causes.

HD duration was defined as the difference between the first HD date and the last follow up date of HD patient.

The factors that may affect SCD are hypertension, diabetes mellitus, and atrial fibrillation (AF). Hypertension was defined as systolic blood pressure ≥140 mmHg or diastolic blood pressure ≥90 mmHg [18]. Diabetes mellitus was defined as fasting blood glucose levels ≥126 mg/dL (fasting: no caloric intake for at least 8 h), hemoglobin A1c levels ≥6.5%, or a previous diagnosis [19]. AF was defined as a documented supraventricular tachyarrhythmia with uncoordinated atrial activation and consequently, ineffective atrial contraction [20].

### 2.4. ECG Analysis

ECG was performed for all participants using an ECG HP-Page-Writer TC 30 machine (filter range 0.5–150 Hz, AC filter 60 Hz, 25 mm/s, 10 mm/mV). We measured the PR interval, which extends from the beginning of the P wave until the beginning of the QRS complex. QRS duration was defined as the time from the first downward deflection after the P wave to the second downward deflection after the R wave. The QT interval was measured from the onset of the QRS complex to end of the T wave that returned to the TP baseline. Every QT interval was corrected for the patient heart rate using Bazett’s formula: QTc = QT√RR (in milliseconds [ms]) [20]. The QTpe interval of each lead was measured from the maximum amplitude of the T wave to the end of the T wave that meets the isoelectric line [20]. QTc dispersion was defined as the difference between the maximum and minimum QTc intervals in 12 leads [20]. The initial ECG was performed when the patient was about to start HD, and the last ECG performed before cardiac death or the end of follow-up in HD patients was considered as the final ECG. All ECG changes were calculated as “ECG parameter before HD” minus “ECG parameter after HD,” and median values were used for analysis. ECG measurements were performed manually with the aid of software (Cardio Calipers, version 3.3, Iconico, Inc., New York, NY, USA) by two researchers blinded to patient information.

### 2.5. Patient Follow-Up

Patients underwent regular check-ups at least twice a week in an open-patient clinic. At each visit, the medical history of the patient was recorded, physical and laboratory examinations were performed, including examinations for infection, drug history, and electrolyte disturbance. We performed ECG at least once in 3 months.

### 2.6. Statistical Analysis

Continuous variables are reported as means ± standard deviations and were compared using *t*-tests. Categorical variables are reported as numbers and percentages and were compared between groups using χ^2^ tests and Fisher’s exact tests. ECG changes before and after HD were compared between groups using a linear mixed model (LMM). SCD risk is presented as a hazard ratio (HR), calculated using a Cox proportional hazards model. The cut-off values of the QTpe intervals in each lead used in Cox analysis were established using receiver operating characteristic (ROC) curves. For additional analyses, propensity score matching was used to reduce the effects of age and sex. All statistical analyses were conducted using the SPSS software, version 25.0 (IBM, Armonk, NY, USA); *p*-values <0.05 were considered significant. Propensity score matching was performed using the R program, version 3.5.3 for Windows.

## 3. Results

### 3.1. Baseline Characteristics

During the study period (mean follow-up duration 63.6 ± 53 months), a total of 678 subjects who underwent serial ECG before and after HD were enrolled and followed-up, among whom 291 died. Infection (*n* = 128, 44%) was the leading cause of death, followed by cardiovascular diseases (*n* = 71, 24.4%), cancer (*n* = 33, 11.3%), intracerebral hemorrhage (*n* = 26, 8.9%), major bleeding (*n* = 13, 4.5%), and others (*n* = 20, 6.9%). Among deaths from cardiovascular diseases, 39 (54.9%) involved SCD (Figure 1A).

Considering survivors and subjects with SCD, the prevalence of SCD was 9−13.6% according to the HD period (Figure 1B). Table 1 shows the baseline characteristics of the subjects, excluding those who died of causes other than SCD (survivors without SCD, *n* = 387; those who died due to SCD, *n* = 39). No significant differences between these two groups were found in sex, body surface area, body mass index, presence of hypertension, mean HD duration, or mean sodium, potassium, or calcium levels measured after HD. However, age (69.2 ± 12.5 vs. 59.8 ± 13.7 years, *p* < 0.001) and the proportion of subjects with diabetes mellitus (78.9% vs. 53.6%, *p* = 0.003) were significantly higher in the SCD group. Regarding TTE findings, the EF was significantly lower in the SCD group than in the survivor group both before and after HD (pre-HD TTE: 54.8 ± 12.4 vs. 59.4 ± 11.3, *p* = 0.037; post-HD TTE: 46 ± 11.8 vs. 58.4 ± 11.6, *p* < 0.001), while the E/E′ ratio was significantly higher in the SCD group than in the survivor group (pre-HD TTE: 19.4 ± 7.7 vs. 15.6 ± 6.6, *p* = 0.011; post-HD TTE: 22.9 ± 9.4 vs. 17.2 ± 7.9, *p* = 0.018). Other parameters, such as left ventricle end-diastolic dimension and left ventricle end-systolic dimension, showed no significant differences.

### 3.2. ECG Findings in the SCD Group

Table 2 and Table 3 show the analysis of ECG findings before and after HD. In the pre-HD ECG analysis, SCD patients had a significantly longer QTpe interval at all leads (II, III, aVF, and V1-6 leads, *p* < 0.001) and a longer QRS duration than survivors (100.6 ± 14.9 vs. 92.6 ± 14.0 ms, *p* = 0.015). The proportion of patients with AF was significantly higher in the SCD group (11.1% vs. 1.9%, *p* = 0.028). Other variables including heart rate, PR interval, QT interval, QTc interval, and dispersion of QT did not show significant differences between the two groups. However, interestingly, in the post-HD ECG analysis, there were no significant differences in any of the variables (QTpe intervals at all leads, QRS duration, or proportion of patients with AF) that had shown differences in the pre-HD ECG analysis. Only heart rate was significantly higher in the SCD group (89.8 ± 22.3 vs. 81.9 ± 15.6 beats/min, *p* = 0.037).

### 3.3. Changes in ECG Parameters before and after HD in the Two Groups

Table 4 and Figure 2 show the median change in each ECG parameter after HD in the two groups. There were no significant differences the changes in QT interval, QTc interval, or dispersion of QT between the two groups. However, the change in QTpe interval after HD was significantly greater in the SCD group than in survivors. Furthermore, in the LMM analysis, the SCD group showed a significantly larger change in the QTpe intervals of the inferior, anterior, and lateral leads after HD than the survivor group (Figure 2, Appendix A; all leads, *p* < 0.001).

### 3.4. ECG Parameters as Predictors of SCD

ROC curve analysis showed the ability of the pre-HD QTpe interval (at all leads) to predict SCD (Appendix A). In particular, the area under the curve of the QTpe interval at the V2 lead was 0.830 (95% confidence interval [CI], 0.741–0.919). The optimal cut-off value of 148.1 ms for the QTpe interval at the V2 lead was determined using ROC curve analysis. Indeed, in multivariable Cox regression analysis, a QTpe interval at the V2 lead >148.1 ms was significantly associated with SCD after adjusting factors including age, sex, diabetes mellitus, AF and EF before HD (HR: 33.793, CI: 4.446–256.845) (Table 5).

### 3.5. Additional Analyses

Since there was a significant difference in age between survivors and SCD patients, additional analyses were performed using propensity score matching to reduce the effects of age and sex (Appendix A). After age- and sex-matching, SCD patients showed a significantly longer pre-HD QTpe interval at all leads (II, III, aVF, and V1-6 leads, *p* < 0.001) than survivors (Appendix A).

Moreover, in an additional analysis based on cardiovascular mortality including SCD, acute coronary diseases, and heart failure, 378 survivors and 71 patients who died of CVD were compared. Detailed results are described in the Appendix A. Similar to the SCD analysis, the additional CVD analysis showed that a longer pre-HD QTpe interval was an important predictor of CVD mortality. Moreover, the changes in some ECG parameters after HD were also significantly greater in the CVD mortality group than in the survivor group (QTpe intervals at the II-V6 leads, *p* < 0.001, Appendix A). The QTc interval was significantly prolonged after HD in the CVD mortality group compared to that in the survivor group (466.9 ± 33.7 vs. 451.8 ± 28.8 ms, *p* = 0.001).

Lastly, to determine whether there was a difference according to the duration of the HD, we studied additional analysis. Both patients who received HD less than 1 year and those who received HD older than 1 year showed similar results to analysis which were conducted with all patients (Appendix A).

## 4. Discussion

In this study, we found that SCD developed in 9−13.6% of HD patients, and these patients showed a significantly longer QTpe interval and QRS duration at all leads before HD initiation. Moreover, ECG changes after HD were significantly greater in the SCD group than in the survivor group.

### 4.1. Clinical Implications of the QTpe Interval in This Study and Previous Studies

SCD in ESRD is a major cause of mortality, and previous studies have reported various risk factors including age, poor nutrition, decreased heart function, and inflammation [2,21]. Among ECG parameters, increased QTpe interval is known to be associated with sudden cardiac death [22]. Our study showed a significantly longer QTpe interval on the pre-HD ECG in the SCD group, similar to this previous study. However, the QTpe interval was shortened after HD in our study. The peak of the T wave represents the end of the repolarization of epicardial cells, while its end represents the end of the repolarization of endocardial cells. Therefore, the interval from the peak to the end of the T wave is closely related to the global transmural dispersion of repolarization (TDR) in the ventricular myocardium [23]. A prolonged QTpe interval may induce an extended vulnerable period and given the right conditions, could increase the risk of ventricular arrhythmogenesis [24]. Moreover, various studies have reported that QTpe interval prolongation is correlated with arrhythmic instability and can be an index of ventricular repolarization [24,25,26,27]. Thus, the QTpe interval predicts ventricular tachycardia/ventricular fibrillation and death, as its increase has been correlated with the risk of ventricular tachyarrhythmia and mortality [28]. Considered the aforementioned results, a longer QTpe interval could be considered a risk factor for SCD.

### 4.2. Comparison with Other Studies: An Explanatory Hypothesis

In our study, the QTpe interval after HD in both groups was close to normal [29], with no differences between the two groups. Our hypothesis to explain these findings is as follows. Chronic kidney disease frequently causes cardiac remodeling with hypertrophy, capillary loss, and fibrosis [30]. Progression of myocardial fibrosis could lead to increased dispersion of repolarization and an increased QTpe interval on ECG [31]. In other words, patients with longer QTpe intervals are more likely to have undergone cardiac remodeling. HD ameliorates the factors causing cardiac remodeling, including volume overload, removes uremic toxins, and corrects electrolyte imbalance [30,32], and this might shorten the QTpe interval. A recent study reported no differences in QTpe intervals between the SCD and survivor groups among HD patients, similar to our results [33]. Another possibility is that a greater change in TDR due to HD resulted in vulnerability to ventricular arrhythmogenesis in a situation involving cardiac remodeling caused by chronic kidney disease. HD patients are always exposed to rapid changes in blood pressure, volume, and electrical conductance of the myocardium when receiving HD. A prolonged QTpe interval before HD suggests a cardiac structural defect, and a defective structure may be more strongly affected by rapid hemodynamic changes. Indeed, in our study, the SCD group showed decreased EF and increased E/E’ on TTE both before and after HD.

### 4.3. Clinical Implications of the QTc Interval and QT Dispersion in This Study and Previous Studies

Several studies have reported an increased QTc interval and increased QT dispersion on ECG in various populations, including ESRD patients, as risk factors for SCD [12,13,34]. However, in our study, the QTc interval was not prolonged compared to its normal value before or after HD [35]. QT dispersion, as in other studies, was >50 ms both before and after HD; hence, it was higher than normal [14,15] but was not different between the SCD and survivor groups. It may be conjectured that previous studies focused on cardiovascular mortality rather than SCD, potentially explaining the difference with our results. In our additional analysis of cardiovascular mortality, the pre-HD QTc interval and QT dispersion were significantly different between the survivor and cardiovascular mortality groups (Appendix A). Although various studies have reported that the QTc interval and QT dispersion are associated with cardiovascular mortality in ESRD patients, a clear association between QT dispersion and SCD in the ESRD population has not yet been observed [11].

### 4.4. Limitations

Our study has several limitations. First, although we enrolled a large sample of patients from three tertiary hospitals, our study population was entirely Asian. Therefore, our findings may not be generalizable to other ethnicities. Second, our study design was retrospective, and some patients’ mortality data were acquired via telephone interviews, which might have led to some bias. However, most ESRD patients with HD underwent regular check-ups at the open-patient clinic, suggesting that any bias would likely be small. Furthermore, we performed additional analyses to reduce the effects of age and sex using propensity score matching. However, some data including vascular access and all serial ECG for each patient was not collected. Further, since enrolled patients were extracted from the registry, some patients might not be included. Therefore, it is necessary to pay attention to the interpretation of the results. Third, we could not confirm our hypothesis about myocardial fibrosis in ESRD patients with HD presented in the Discussion. Fibrosis of the myocardium could be confirmed using biomarkers (tumor necrosis factor alpha, matrix metalloproteinase-2, and cytokines) or through magnetic resonance imaging. Lastly we only compared the initial ECG before start HD with the last ECG performed before cardiac death or the end of follow-up in HD patients. Therefore, we could not find ECG parameters change over time the longer patients stayed on HD in this study. Therefore, further studies are needed to confirm and improve on our results.

## 5. Conclusions

Our results demonstrate that ECG parameters before and after HD may help in predicting SCD in ESRD patients. The QTpe interval at all leads and QRS duration were shortened after HD. Patients with a longer QTpe interval before HD and large ECG changes after HD might have a relatively higher SCD risk.

## Figures and Tables

**Figure 1 jcm-10-01933-f001:**
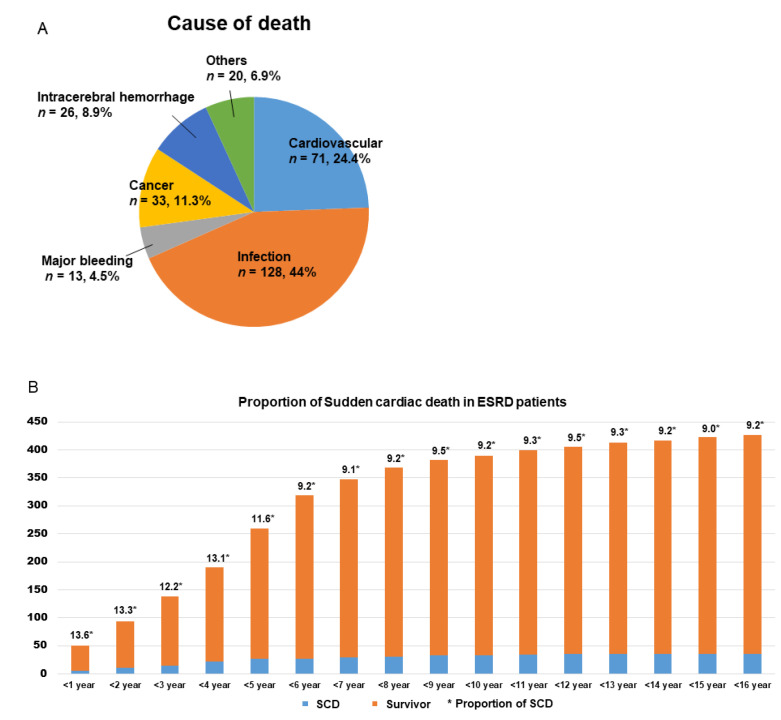
Cause of death and proportion of sudden cardiac death in end-stage kidney disease (ESRD) patients. (**A**) Cause of death in ESRD patients. (**B**) Proportion of sudden cardiac death in ESRD patients.

**Figure 2 jcm-10-01933-f002:**
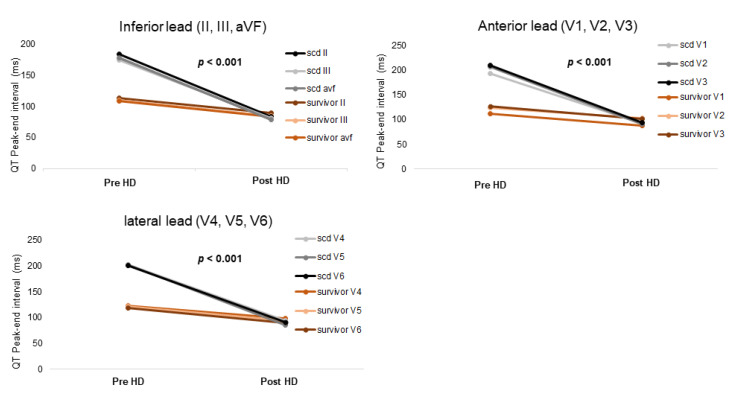
Comparison of electrocardiography (ECG) change before and after hemodialysis (HD) using a linear mixed model. The change in QT peak-to-end (QTpe) interval before and after HD in the sudden cardiac death (SCD) group and the change in QTpe interval before and after HD in the survivor group were statistically significant. The SCD group showed a significantly larger change in the QTpe intervals of the inferior, anterior, and lateral leads after HD than the survivor group.

**Table 1 jcm-10-01933-t001:** Baseline characteristics of the patients.

Variable	Survivor (*n* = 387)	SCD (*n* = 39)	*p*-Value
Age (years)	59.8 ± 13.7	69.2 ± 12.5	<0.001
Male (*n*, %)	189 (48.8)	25 (64.1)	0.069
Body surface area (m^2^)	1.64 ± 0.19	1.66 ± 0.22	0.571
Body mass index (kg/m^2^)	23.0 ± 3.7	22.8 ± 3.9	0.78
Hypertension (*n*, %)	349 (90.2)	33 (84.6)	0.387
Diabetes mellitus (*n*, %)	206 (53.2)	30 (76.9)	0.003
Duration of HD (months)	64.5 ± 53.0	54.9 ± 52.8	0.294
Echocardiography before HD			
Pre-HD LVEF (%)	59.4 ± 11.3	54.8 ± 12.4	0.037
Pre-HD LVEDD (mm)	51.9 ± 6.6	53.9 ± 10.1	0.367
Pre-HD LVESD (mm)	35.5 ± 7.3	38.1 ± 11.2	0.329
Pre-HD E/E′	15.6 ± 6.6	19.4 ± 7.7	0.011
Echocardiography after HD			
Post-HD- HD LVEF (%)	58.4 ± 11.6	46 ± 11.8	<0.001
Post-HD LVEDD (mm)	51.2 ± 6.0	52.7 ± 10.5	0.493
Post-HD LVESD (mm)	35.3 ± 7.1	39.9 ± 10.7	0.051
Post-HD E/E′	17.2 ± 7.9	22.9 ± 9.4	0.018
Laboratory findings after HD			
Hb (g/dL)	10.5 ± 1.4	9.3 ± 1.5	<0.001
Hct (%)	32.6 ± 7.8	28.2 ± 4.8	0.017
BUN (mg/dL)	54.8 ± 21.9	58.5 ± 30.7	0.611
Cr (mg/dL)	8.4 ± 3.2	6.8 ± 3.7	0.039
eGFR (EPI)	6.5 ± 5.6	10.0 ± 8.4	0.091
Sodium (mEq/L)	137 ± 9.8	137.6 ± 5.4	0.764
Potassium (mEq/L)	4.9 ± 0.8	4.8 ± 1.3	0.642
T.Calcium (mg/dL)	8.9 ± 1.0	8.6 ± 0.7	0.352

Values are expressed as *n* (%) or means (standard deviations) unless otherwise indicated; SCD, sudden cardiac death; HD, hemodialysis; LV, left ventricle; EF, ejected fraction; LVEDD, left ventricle end-diastolic dimension; LVESD, left ventricle end-systolic dimension; E/E′, E: mitral peak velocity of early filling, E′: early diastolic mitral annular velocity; Hb, hemoglobin; Hct, hematocrit; BUN, blood urea nitrogen; Cr, creatinine; eGFR, estimated glomerular filtration rate; T.Calcium, Total calcium.

**Table 2 jcm-10-01933-t002:** Baseline ECG findings before hemodialysis.

Variable	Survivor (*n* = 387)	SCD (*n* = 39)	*p*-Value
Atrial fibrillation (*n*, %)	5 (1.3)	3 (7.7)	0.028
Heart rate (/min)	76.2 ± 14.6	80.6 ± 16.7	0.141
PR interval (ms)	168.9 ± 27.6	175.2 ± 37.9	0.377
QRS duration (ms)	92.6 ± 14.0	100.6 ± 14.9	0.015
QT interval (ms)	405.5 ± 47.9	395.6 ± 49.0	0.307
QTc interval (ms)	451.8 ± 28.8	456.2 ± 29.9	0.449
Dispersion of QT (ms)	55.5 ± 32.4	67.9 ± 22.7	0.052
QT Peak-end interval (ms)			
II	113.4 ± 50.0	183.9 ± 67.4	<0.001
III	110.0 ± 69.2	174.8 ± 72.3	<0.001
aVF	108.4 ± 48.4	178.4 ± 70.8	<0.001
V1	112.0 ± 49.6	192.5 ± 70.5	<0.001
V2	123.9 ± 51.5	206.8 ± 67.7	<0.001
V3	126.3 ± 53.3	209.7 ± 75.9	<0.001
V4	122.2 ± 54.5	201.8 ± 71.8	<0.001
V5	121.1 ± 53.6	202.1 ± 76.7	<0.001
V6	118.5 ± 54.4	200.3 ± 80.8	<0.001

Values are expressed as *n* (%) or means (standard deviations) unless otherwise indicated; SCD, sudden cardiac death; ECG, electrocardiography.

**Table 3 jcm-10-01933-t003:** Baseline ECG findings after hemodialysis.

Variable	Survivor (*n* = 387)	SCD (*n* = 39)	*p*-Value
Atrial fibrillation (*n*, %)	12 (3.1)	3 (7.7)	0.138
Heart rate (/min)	81.9 ± 15.6	89.8 ± 22.3	0.037
PR interval (ms)	171.2 ± 31.8	151.0 ± 44.6	0.106
QRS duration (ms)	95.2 ± 26.1	105.6 ± 33.0	0.099
QT interval (ms)	409.0 ± 51.4	396.9 ± 62.7	0.279
QTc interval (ms)	459.6 ± 46.8	464.1 ± 47.3	0.601
Dispersion of QT (ms)	52.0 ± 21	66.6 ± 43.8	0.075
QT Peak-end interval (ms)			
II	89.1 ± 37.0	83.3 ± 37.1	0.419
III	84.2 ± 52.5	81.5 ± 35.3	0.781
aVF	83.3 ± 20.3	78.4 ± 31.9	0.413
V1	88.3 ± 22.9	92.0 ± 22.0	0.407
V2	102.1 ± 50.5	90.4 ± 21.4	0.232
V3	102.0 ± 20.0	94.2 ± 24.8	0.096
V4	98.4 ± 26.8	93.7 ± 28.3	0.382
V5	93.5 ± 21.9	86.0 ± 22.2	0.079
V6	90.0 ± 20.5	91.3 ± 25.8	0.802

Values are expressed as *n* (%) or means (standard deviations) unless otherwise indicated; SCD, sudden cardiac death; ECG, electrocardiography.

**Table 4 jcm-10-01933-t004:** Changes in ECG parameters after HD (ECG before HD-ECG after HD).

Variable	Survivor (*n* = 387)	SCD (*n* = 39)	*p*-Value
QT interval (ms)	−6 (−36, 26)	38 (−25, 84)	0.963
QTc interval (ms)	−8.5 (−32, 12)	1 (−13.5, 58)	0.937
Dispersion of QT (ms)	4.3 (−13.9, 27.8)	32 (1, 48)	0.02
QT Peak-end interval (ms)			
II	8 (−9.4, −71.2)	112 (82, 128)	<0.001
III	12 (−8, 69.4)	96 (66, 140)	<0.001
aVF	12 (−8, 61.8)	132 (96, 142)	<0.001
V1	8 (−12, 64.4)	140 (98.0−150)	<0.001
V2	12 (−8, 63)	140 (94, 168)	<0.001
V3	8.7 (−8, 70.2)	152 (114, 186)	<0.001
V4	8 (−10, 64.7)	140 (80, 164)	<0.001
V5	12 (−9.2, 96.6)	160 (142, 176)	<0.001
V6	12 (−4.2, 66.2)	152 (90, 174)	<0.001

Values are expressed as medians (interquartile range); ECG, electrocardiography; HD, hemodialysis; SCD, sudden cardiac death.

**Table 5 jcm-10-01933-t005:** Cox analysis of the risk of sudden cardiac death predicted by ECG parameters.

	Hazard Ratio	95% Confidence Interval	*p*-Value
Age	1.051	1.005–1.100	0.030
Sex, male	1.846	0.623–5.470	0.269
Diabetes mellitus	3.276	0.719–14.936	0.125
Atrial fibrillation before hemodialysis	8.061	1.886–34.449	0.005
Ejection fraction in echocardiography before hemodialysis	0.975	0.934–1.019	0.259
QTpe interval at the V2 lead > 148.1 ms before hemodialysis	33.793	4.446–256.845	0.001

QTpe: QT peak-to-end.

## Data Availability

The datasets generated and/or analyzed during the current study are not publicly available due to our IRB policy but are available from the corresponding author upon reasonable request.

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
