# Peer review of "Clinical Associations between Serial Electrocardiography Measurements and Sudden Cardiac Death in Patients with End-Stage Renal Disease Undergoing Hemodialysis"

_jcm, 2021, doi:10.3390/jcm10091933_

Round 1

Reviewer 1 Report

In this study, Lee et al. investigate the association between various ECG measurements and SCD in patients initiating HD. The major findings of the study include a significant association between pre-HD QTpe interval (though not post-HD) and the absence of an association between QTc and SCD. Overall, SCD is an important clinical topic for nephrologists and efforts to identify potentially modifiable risk factors are laudable. I have a few comments listed below: 

Major:

- Was the vascular access for each patient known? If so, this should be reported as well in Table 1.

- For the evaluation of parameters as predictors of SCD, was any internal validation strategy applied to limit the risk of overfitting since the optimal cutpoint for QTpe interval was derived from the same dataset?

- Since the incidence and mechanisms of death in the first few months of dialysis differ from other time periods after dialysis initiation, I wonder if any of these associations differ when limited to analysis of SCD cases in this time period. While the number of cases is likely small, perhaps an additional secondary analysis focusing on this time period would be useful.

Minor:

- Abstract: Should read “multi-center study.”

- Discussion, Section “Clinical implications of the QTpe interval in this study and previous studies”, line 10: Replace the comma with a period at the end of the study.

- Discussion, Section “Comparison with other studies: an explanatory hypothesis”, last line: SCD group showed an increased not decreased E/E’ on TTE.

- The figure legend for Figure S1B  is copied from Figure 2 of the main manuscript and not the actual description of Figure S1B.

- Which criteria were applied to consider a metabolic disorder “uncontrolled.”

Author Response

Dear editor

Manuscript ID: cm-1173031

Manuscript Title: Clinical associations between serial electrocardiography measurements and sudden cardiac death in patients with end-stage renal disease undergoing hemodialysis

We would like to thank all of the editors and reviewers for helping us make a better revision. We revised our manuscript according to the comments and recommendations of the reviewers. We highlighted all changes in the revised manuscript in blue letters. Here we include a separate itemized series of responses to the comments of the reviewers.

- Response to Reviewers -

Reviewer #1

In this study, Lee et al. investigate the association between various ECG measurements and SCD in patients initiating HD. The major findings of the study include a significant association between pre-HD QTpe interval (though not post-HD) and the absence of an association between QTc and SCD. Overall, SCD is an important clinical topic for nephrologists and efforts to identify potentially modifiable risk factors are laudable. I have a few comments listed below:

Major:

  1. Was the vascular access for each patient known? If so, this should be reported as well in Table 1.

à Thank you for your comment. Unfortunately, we did not plan to acquire vascular information during the initial plan. Therefore, we did not acquire the vascular access data of patients. We have added the following to the limitation section: However, some data including vascular access and all serial ECG for each patient was not collected. Therefore, it is necessary to pay attention to the interpretation of the results. (page 10)

  1. For the evaluation of parameters as predictors of SCD, was any internal validation strategy applied to limit the risk of overfitting since the optimal cutpoint for QTpe interval was derived from the same dataset?

à Thank you for detailed review. At the time of the initial analysis, because the aim of this study was not to develop a predictive model about SCD, but to find the association between the development of SCD and change in ECG parameter, we did not conduct internal validation about cutpoint of QTpe interval. After reviewer's comment, we tried internal validation and sought advice form statistics expert. Unfortunately, we have not been able to do internal validation, because current data set did not have enough SCD patients for internal validation. It is believed that additional data will be needed for validation, therefore, further study is necessary. we described necessity of further study in limitation section. (page 10)

  1. Since the incidence and mechanisms of death in the first few months of dialysis differ from other time periods after dialysis initiation, I wonder if any of these associations differ when limited to analysis of SCD cases in this time period. While the number of cases is likely small, perhaps an additional secondary analysis focusing on this time period would be useful.

à We re-analyzed according to your recommendation. Patients were divided according to duration of hemodialysis. Both patients who received hemodialysis less than 1 year and those who received hemodialysis older than 1 year showed similar results. There were 8 patients with SCD within 1 year after hemodialysis. Those patients were significantly longer QTpe intervals in all leads and were shortened after hemodialysis. The same result was shown for patients who received hemodialysis older than 1 year. We added the analysis results of patients who received hemodialysis older than 1 year in supplementary data. And we have added the following sentence to the Additional analyses sections: Lastly, to determine whether there was a difference according to the duration of the HD, we studied additional analysis. Both patients who received HD less than 1 year and those who received HD older than 1 year showed similar results to analysis which were conducted with all patients (Supplement Tables S9-12).  (page 9)

Minor:

  1. Abstract: Should read “multi-center study.”

à We agree with the reviewer’s opinion. We edited to multi-center study from multi-center. (page 1)

  1. Discussion, Section “Clinical implications of the QTpe interval in this study and previous studies”, line 10: Replace the comma with a period at the end of the study.

à We agree with the reviewer’s opinion. We edited. (page 9)

  1. Discussion, Section “Comparison with other studies: an explanatory hypothesis”, last line: SCD group showed an increased not decreased E/E’ on TTE.

à Thank you for review. We edited to “decreased EF and increased E/E’” from “decreased EF and increased E/E’”. (page 10) 

  1. The figure legend for Figure S1B is copied from Figure 2 of the main manuscript and not the actual description of Figure S1B.

à We changed legend of figure 1A, figure 2B in supplementary data and it inserted in results section in manuscript. We have added the following to results section:

Figure 1 Cause of death and proportion of sudden cardiac death in ESRD patients. (A) Cause of death in ESRD patients. (B) Proportion of sudden cardiac death in ESRD patients.

  1. Which criteria were applied to consider a metabolic disorder “uncontrolled.”

à In this study, uncontrolled metabolic disorder means that diseases are difficult to control due to genetic problems such as homocystinuria, glycogen storage diseases, and mitochondrial disorders. As it may cause confusion, we edited to the following sentence in method section: any uncontrolled metabolic disorders such as homocystinuria, glycogen storage diseases, and mitochondrial disorders. (page 2)

Reviewer 2 Report

The authors describe a cohort of patients in three corean dialysis clinics and assessses them for sudden cardiac death and potential predictors of SCD. For this they especially looked at subtle differences in ECGs between survivors and patients who died from sudden cardiac death.

Methods:

  1. During which period of time where the patients enrolled and how long was the follow up?
  2. Was every patient that started hemodialysis included? How many patients were excluded and due to which reasons?

Results:

Table 1: baseline characteristics

  • What time was chosen as mean HD duration? In the methods section you stated that you included patients when they started hemodialysis? How can mean dialysis time be around 5 years?

Figure 1: Comparison of ECG change before and after HD using a linear mixed model.

  • The legend is not sufficient to explain the figures. What do the lines in the graphs stand for? Individual patients? Why were these patients chosen? What is depicted on the y axis?

figure 2 in the supplement

  • The legend for figure 2 in the supplement seems not correct. It states “Comparison of ECG changes before and after HD in survivors and SCD patients using a linear mixing model” however the figure shows the proportion of sudden cardiac death in ESRD patients. The legend should be adjusted to explant the data presented.

Other major points:

Why where the other significant differences in the baseline characteristics (e.g. changes in echocardiographic parameters, the presence of atrial fibrillation or hemoglobin levels) not included in the propensity score matching? A smaller ejection fraction in the SCD group and a higher decline in EF during hemodialysis is most likely also a factor influencing the risk of cardiac death and probably even more important than changes in the ecg. The same might account for atrial fibrillation as a risk factor or a lower hemoglobin level/hematocrit, indicating possible sicker patients. The influence of this parameters on cardiac deaths in relation to ecg changes does not become clear.

Were differences in ECG between survivors and patients dying from SCD only assessed at baseline e.g. when they began dialysis or were the some differences also noticeable at later time points? Did the assessed ECG parameters change over time the longer patients stayed on dialysis?

Another import point is, that it is clear to me, why some of the primary data that is referred to in the text was placed in the supplements. For the reader this means that one has to switch constantly between the original paper and the supplements in order to understand the main aspects of the paper. The authors should choose better which data needs to be shown in the paper and which data only supplements the main message in order to improve readability of their work.

Author Response

Dear editor

Manuscript ID: cm-1173031

Manuscript Title: Clinical associations between serial electrocardiography measurements and sudden cardiac death in patients with end-stage renal disease undergoing hemodialysis

We would like to thank all of the editors and reviewers for helping us make a better revision. We revised our manuscript according to the comments and recommendations of the reviewers. We highlighted all changes in the revised manuscript in blue letters. Here we include a separate itemized series of responses to the comments of the reviewers.

- Response to Reviewers -

Reviewer #2

The authors describe a cohort of patients in three corean dialysis clinics and assessses them for sudden cardiac death and potential predictors of SCD. For this they especially looked at subtle differences in ECGs between survivors and patients who died from sudden cardiac death.

Methods:

  1. During which period of time where the patients enrolled and how long was the follow up?

à Thank you for review. We enrolled 678 patients who underwent HD retrospectively, from November 1986 to September 2015 at three universities medical centers (Ewha Womans University Medical Center, National Health Insurance Service Ilsan Hospital, Yonsei University Medical Center) in Korea. The last follow up date is 26-November 2016. Mean follow up month is 53.4 ± 53.8 and median follow up month is 40.6 (IQR 15.8-65.4).

We have added the following to the methods section: This study enrolled patients who started HD in each hospital from November 1986 to September 2015. And the last follow up date was 26-November 2016. (page 2)

  1. Was every patient that started hemodialysis included? How many patients were excluded and due to which reasons?

à Enrolled patients were retrospectively extracted using CDR data registry. Since we extracted data by entering several conditions such as including and excluding criteria in the CDR, we could not number of every hemodialysis patient. We added these contents in limitation section: And since enrolled patients were extracted from the registry, some patients might not be included. (page 10)

Results:

Table 1: baseline characteristics

  1. What time was chosen as mean HD duration? In the methods section you stated that you included patients when they started hemodialysis? How can mean dialysis time be around 5 years?

à We have added the following to the methods section: This study enrolled patients who started HD in each hospital from November 1986 to September 2015. And the last follow up date was 26-November 2016. (page 2)

The Mean HD duration is the average of the difference between the first HD date and the last follow up date for enrolled patients. To prevent confusion, we have added the following to the definition section: HD duration was defined the difference between the first HD date and the last follow up date of HD patient. (page 3)

Figure 1: Comparison of ECG change before and after HD using a linear mixed model.

  1. The legend is not sufficient to explain the figures. What do the lines in the graphs stand for? Individual patients? Why were these patients chosen? What is depicted on the y axis?

à We agree with the reviewer’s opinion. The lines in the graphs show change median change in each ECG parameter between before and after HD in the two groups. These not mean specific individual patient but all enrolled patients. Y axis means QT Peak-end interval (ms). We modified the figure 1, and figure S2, S3 by inserting the y axis meaning. And we have added the following to the figure 1 (we modified to figure 2) legend: The change in QTpe interval before and after HD in the SCD group and the change in QTpe interval before and after HD in the survivor group were statistically significant. The SCD group showed a significantly larger change in the QTpe intervals of the inferior, anterior, and lateral leads after HD than the survivor group. (page 8)

figure 2 in the supplement.

  1. The legend for figure 2 in the supplement seems not correct. It states “Comparison of ECG changes before and after HD in survivors and SCD patients using a linear mixing model” however the figure shows the proportion of sudden cardiac death in ESRD patients. The legend should be adjusted to explant the data presented.

à Thank you for review. We edited all figure legends to fit the order. And figure S2 modified to figure S1.

Other major points:

  1. Why where the other significant differences in the baseline characteristics (e.g. changes in echocardiographic parameters, the presence of atrial fibrillation or hemoglobin levels) not included in the propensity score matching? A smaller ejection fraction in the SCD group and a higher decline in EF during hemodialysis is most likely also a factor influencing the risk of cardiac death and probably even more important than changes in the ecg. The same might account for atrial fibrillation as a risk factor or a lower hemoglobin level/hematocrit, indicating possible sicker patients. The influence of this parameters on cardiac deaths in relation to ecg changes does not become clear.

à We reanalyzed according to reviewer’s opinion. We conducted cox analysis of the risk of sudden cardiac death predicted by ECG parameters including age, sex, DM, AF, EF, and QTpe interval at the V2 lead (> 148.1 ms) before HD. The results of analysis showed that a QTpe interval at the V2 lead >148.1 ms was significantly associated with SCD after adjusting factors such as age, sex, DM, AF and EF (HR: 38.134, CI: 5.039–288.617). In the previous analysis, there were 39 patients with SCD and the number of patients with SCD for analysis including many variables was small. Therefore, we limited the number of variables.

In accordance with your recommendations, we revised the ECG parameters as predictors of SCD section and supplement table 2. This supplementary table's title was change to table 5 and was inserted into the results section of the manuscript.

(Cox regression analysis, a QTpe interval at the V2 lead >148.1 ms was significantly asso-ciated with SCD after adjusting factors including age, sex, diabetes mellitus, AF and EF before HD (HR: 33.793, CI: 4.446–256.845) (Table 5). (page 8)

  1. Were differences in ECG between survivors and patients dying from SCD only assessed at baseline e.g. when they began dialysis or were the some differences also noticeable at later time points? Did the assessed ECG parameters change over time the longer patients stayed on dialysis?

à Unfortunately, we could not obtain all serial ECG data of each patient. The initial ECG was performed when the patient was about to start HD, and the last ECG performed before cardiac death or the end of follow-up in HD patients was considered as the final ECG. We could only compare the ECG before and after hemodialysis. Therefore, we could not find ECG parameters change over time the longer patients stayed on dialysis. We have added the following to the limitation section: Lastly we only compared the initial ECG before start HD with the last ECG performed before cardiac death or the end of follow-up in HD patients. Therefore, we could not find ECG parameters change over time the longer patients stayed on HD in this study. (page 9)

  1. Another import point is, that it is clear to me, why some of the primary data that is referred to in the text was placed in the supplements. For the reader this means that one has to switch constantly between the original paper and the supplements in order to understand the main aspects of the paper. The authors should choose better which data needs to be shown in the paper and which data only supplements the main message in order to improve readability of their work.

à We agree reviewer’s opinion. All the results of the primary analysis were located in the manuscript. Supplementary figure 1 was modified to figure 1 and inserted into result section (Baseline characteristics). And supplementary table 2 was modified to table 5 and inserted result section (ECG parameters as predictors of SCD). Additional analysis results were collected in the supplementary data. (page 4, 5, 8)
